# Counterfactual Causal Inference in Natural Language with Large Language Models

## Abstract

Causal structure discovery methods are commonly applied to structured data where the causal variables are known and where statistical testing can be used to assess the causal relationships. By contrast, recovering a causal structure from unstructured natural language data such as news articles contains numerous challenges due to the absence of known variables or counterfactual data to estimate the causal links. Large Language Models (LLMs) have shown promising results in this direction but also exhibit limitations. This work investigates LLM's abilities to build causal graphs from text documents and perform counterfactual causal inference. We propose an end-to-end causal structure discovery and causal inference method from natural language: we first use an LLM to extract the instantiated causal variables from text data and build a causal graph. We merge causal graphs from multiple data sources to represent the most exhaustive set of causes possible. We then conduct counterfactual inference on the estimated graph. The causal graph conditioning allows reduction of LLM biases and better represents the causal estimands. We use our method to show that the limitations of LLMs in counterfactual causal reasoning come from prediction errors and propose directions to mitigate them. We demonstrate the applicability of our method on real-world news articles.

## 1 Introduction

Recovering the causal structure of events described in single or multiple text sources is an important problem for natural language understanding, analysis, and prediction. In particular, causal structure discovery from news articles can help build causal world models that can be used to understand the causal chains behind events, forecast future events, and create robust automated reasoning agents (Schölkopf et al., 2021; Bareinboim et al., 2022). This problem is not often tackled in natural language processing (NLP) because it requires solving multiple challenges considered open research problems in causality and NLP research. First, the text modality prevents the direct use of traditional structure discovery methods as the set of causal variables is not available and has to be discovered (a problem being recently tackled by the field of causal representation learning) (Schölkopf et al., 2021). Second, real-world events have non-trivial structures prone to latent variables and feedback loops, typically excluded from most causal analyses using Direct Acyclic Graphs (DAGs) (Bongers et al., 2021). Third, causal models provide a means to answer interventional and counterfactual queries, but real-world data prevents the ground truth from being directly accessible. This is the *fundamental problem of causal inference* (Pearl, 2009): only one factual world can be observed. In addition, real-world events are often complex and multi-causal, greatly hindering the possibility of manually annotating a ground truth.

Large Language Models (LLMs) have demonstrated impressive abilities to solve tasks related to these problems, notably for understanding and summarising natural language (Devlin et al., 2019; Brown et al., 2020; Bubeck et al., 2023). Recent work (Jiralerspong et al., 2024) has also shown that LLMs can recover causal structures, although the authors do not apply it to text data. However, this ability is still debated as the LLM's performance can drop significantly when facing unfamiliar settings (Jin et al., 2024). Moreover, while being able to perform some causal reasoning tasks successfully (Melnychuk et al., 2022), notably on commonsense causal reasoning (Kiciman et al., 2023; Zhang et al., 2022), LLMs have shown limitations on tasks that require robust reasoning, such as arithmetic tasks in unfamiliar settings (Wu et al., 2023), or abstract reasoning (Gendron et al., 2024). To explain this behavior difference, (Zecevic et al., 2023) advanced that LLMs cannot discover new

causal relationships but only recall ones already seen during training. Acknowledging this limitation, alternative approaches have been proposed using LLMs to extract causal relationships from text (Rožanec et al., 2023b; Rozanec et al.). However, such approaches do not formally test whether the extracted relationships are causal.

We investigate ways to overcome this restriction and propose an end-to-end causal structure discovery and counterfactual inference method from purely unstructured natural language text data. Our framework is divided into two steps: first, we use an LLM to generate the causal graph associated with a document, i.e., that describes the causal relationships between the events depicted in the text. Optionally, we merge causal graphs from multiple sources using a second LLM. Then, we use the built causal structure to perform an atomic intervention on the sequence of events and infer the consequences in this counterfactual scenario using an LLM (i.e., answer *what if?* questions). We show that our method can effectively extract causal relationships and propose plausible counterfactual worlds from real-world events.

Our contributions can be summarised as follows:

- We propose a method to perform causal structure discovery and counterfactual inference in an end-to-end and explainable way,
- We demonstrate its applicability on real-world events,
- We use our method to disentangle the steps required for counterfactual reasoning and highlight the limitations of LLMs. We show that LLMs can fail even when the full reasoning structure is given and that the bottleneck for performance comes from the prediction step.

Our code is available at this anonymous repository: `https://anonymous.4open.science/r/counterfactual-llm-inference-84BB`

## 2 RELATED WORK

**Causal Structure Discovery with LLMs**  Text data is often unstructured, high-dimensional, and large-scale. Causal variables may not be directly accessible from the text, and the causal relationships are typically vague and rare, with semantic ambiguity complicating analysis. These challenges greatly hinder the usability of traditional causal structure discovery methods and motivate using LLMs for causal structure discovery (Kiciman et al., 2023; Ma, 2024). (Kiciman et al., 2023) have achieved promising results when using LLMs to infer the causal direction between two variables. Nevertheless, there is some evidence that LLMs, in many cases, repeat embedded causal knowledge (Zecevic et al., 2023) and are susceptible to inferring causal relations from the order of two entities mentioned in a text (Joshi et al., 2024). (Jiralerspong et al., 2024) attempts to recover the full causal structure using a breadth-first search on a set of text variables. (Hobbhahn et al., 2022) investigates whether LLMs can identify the cause and the effect between two natural language sentences. This line of work uses the LLM's inner knowledge to discover causal relationships between data points. However, recent work highlighted that LLMs do not conduct proper causal reasoning and mainly rely on domain knowledge and correlations (Zecevic et al., 2023; Jin et al., 2024). This approach differs from another line of work that uses the LLM as an information retrieval engine to extract causal relationships explicitly present in the data. For instance, (Gopalakrishnan et al., 2024) uses LLMs to extract causal relationships from medical texts. Our approach combines both worlds as we use an LLM to retrieve causal relationships from text and then perform causal inference on the extracted model to assess the quality of the causal model. NATURAL (Dhawan et al., 2024) is another method developed concurrently to our work using LLMs to perform causal inference. However, the method is based on the Potential Outcome Framework (POF) and computes the Average Treatment Effect (ATE) of a variable (i.e., outcome) under an intervention (i.e., a treatment). By contrast, our method is based on Structural Causal Models (SCMs) and includes a causal structure discovery step to represent complex causal relationships between observed variables. Our work also focuses on counterfactuals, while this method is suited to answer interventional queries.

**Strategic Foresight**  Strategic Foresight aims to provide a structured approach to gathering information regarding plausible future scenarios and adequately preparing for change. It provides expert insights regarding trends and emerging issues that can be considered for strategic planning and policy-making. As such, it is being increasingly adopted in the public and private sectors (Burt & Nair,

2020; Rosa et al., 2021). Among the most frequently used methods, we find scenario planning (Ebadi et al., 2022), which aims to foresee relevant scenarios based on trends and factors of influence to understand better how actions can influence the future (Wilkinson, 2017). While the value of artificial intelligence for strategic foresight has been recognized, much of the work is still not automated (Reez, 2020; Brandtner & Mates, 2021). Scientific literature reports on using artificial intelligence for information scanning and analysis (Parrish et al., 2019; Brandtner & Mates, 2021), to identify weak signals and trends (Geurts et al., 2022), and extract actions and outcomes that can be mapped to causal decision diagrams (Pratt et al., 2023). More recently, authors have proposed architectures that could automate strategic foresight. Nevertheless, the proposed architecture only considered a signal assessment module without explicit reference to testing causality among the extracted graph relationships (Rozanec et al., 2023; Rožanec et al., 2023a). We aim to bridge this gap by identifying, extracting, and testing causal relationships reported in media news to construct a graph of causal relationships and use such a graph to build plausible future scenarios, providing expert insights for strategic planning and policy-making.

# 3 Counterfactual Inference with Large Language Models

This section describes our proposed method for causal structure discovery and causal inference from text. We build a modified version of the Structural Causal Model (SCM) (Pearl, 2009) to represent the described causal mechanisms. Following the SCM framework, we consider that the mechanisms can be represented as Directed Acyclic Graphs (DAGs). For every document $D$, we construct a DAG $\mathcal{G} = \langle \mathbf{U}, \mathbf{V}, \mathbf{E} \rangle$ where the observed nodes $v \in V$ correspond to general depictions of the events in the text. Two nodes $(V_i, V_j) \in \mathbf{V} \times \mathbf{V}$ are connected by an edge $E_{ij} = (V_i, V_j) \in \mathbf{E}$ if a causal relationship between them is explicitly mentioned in the text data (according to an LLM). We also represent possible exogenous factors $U \in \mathbf{U}$ describing unobserved events having a causal influence on the observations. Nodes and edges also have features. Nodes correspond to causal variables and have a *domain* established during extraction and a *current value* from this domain. Confounders $U$ are not assigned values because they are not observed. As we aim to use an LLM for inference, we also keep attributes in plain natural language: a *description* of the variable and additional *contextual information*. Edges also have a *description* attribute. For example, given this sentence: "The airlines companies have seen their revenues diminishing due to travel restrictions.", we can extract the following causal variables $S$ and $T$, and their relationship: *travel restrictions (S)* $\rightarrow$ *airlines revenues (T)*. Their domains can be, e.g., a boolean for $S$ and a fixed set of categories for $T$ (since we do not have access to the numerical values of the revenues). Their current values are written $S = s_o$ and $T = t_o$. During counterfactual inference, we intervene to modify these values while the other attributes remain unchanged. For performance, we add contextual information to each node, i.e., background knowledge extracted from the text. In this example, the contextual information could be the country where the event takes place. The edge description can be, e.g., "travel restrictions diminish airlines companies revenues".

SCMs typically have a set of mapping functions $\mathcal{F}$ to infer the value of a variable $V$ given its parents $\mathbf{pa}(\cdot)$ (e.g., $V \leftarrow f_V(\mathbf{pa}(V))$). Instead of using a set of predefined functions, we perform causal inference using an LLM. We also use it to compute the prior probability distribution of the confounders $U$. We discuss our method and the implications of using an LLM for inference in Section 3.2. We note a full causal model containing the causal graph $\mathcal{G}$ and all these attributes as $\mathcal{M} = \langle \mathcal{G}, \text{LLM} \rangle$. The instantiated model $\mathcal{M}(D)$ describes the model with the values of each variable extracted from the document $D$. We further note an intervention $do(X = x)$ in this model as $\mathcal{M}_{X=x}(D)$ or $\mathcal{M}(D, do(X = x))$.

Our proposed method is divided into four stages. First, we extract the causal graph from a text document. If multiple documents are provided, we merge their respective causal graphs together. Then, we compute counterfactual worlds from the resulting causal graph. We use these counterfactuals to self-evaluate the causal graph. Section 3.1 describes the causal graph extraction step. Section 3.2 describes the counterfactual inference step. Evaluation is described in Sections 4.3 and 5. Figure 1 illustrates the complete pipeline.

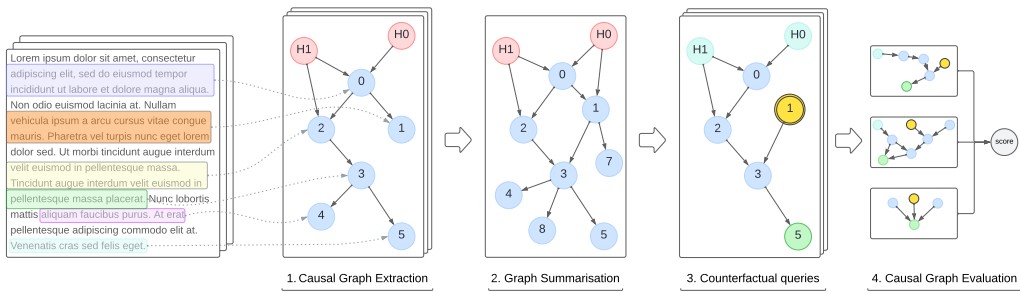

Figure 1: Overview of the proposed framework. (1) An LLM extracts causal variables and their corresponding causal relationships from the input text. (2) Multiple graphs are generated and merged into a single graph if multiple text snippets are given in input. (3) The resulting causal graph is edited using ablation, intervention, and prediction steps to build counterfactual instantiations. The LLM performs inference given the variables' parent values. (4) The LLM self-evaluates the original and counterfactual graphs.

## 3.1 CAUSAL STRUCTURE DISCOVERY FROM NATURAL LANGUAGE

We prompt an LLM to read an input document and return its associated causal graph. The expected graph should contain the full set of observed causal variables, their relationships, and their attributes as described in the introduction of Section 3. We also ask the LLM to estimate the hidden variables affecting the observed events and how they are connected. Estimated hidden variables have the same attributes as the observed ones but do not have an observed value. Other works have studied ways to improve the quality of the generated causal graphs, e.g., via breadth-first search (Jiralerspong et al., 2024). However, to keep our pipeline efficient, we only consider a single forward pass with chain-of-thought prompting (Wei et al., 2022). We find that this simple choice is sufficient for our purpose. This is not surprising as the causal relationships to be found are explicitly described in the data and LLMs have been very successful at information retrieval tasks (Brown et al., 2020; Bubeck et al., 2023; Reid et al., 2024). We prompt the LLM to return its answer in JSON format. Still, it may not always provide an extractable response. To alleviate this issue, we allow the LLM to refine its answer several times if it cannot be parsed automatically. If multiple documents are provided, we merge the causal graphs together. We describe this optional step in Appendix A.

## 3.2 COUNTERFACTUAL CAUSAL INFERENCE

Autoregressive LLMs are inference engines computing the conditional probability of an output token $Y_0$ given an input context $\mathbf{C}$: $P(Y_0|\mathbf{C})$. When used for generation, they construct an output sequence $\mathbf{Y} = [Y_0, \ldots, Y_N]$ by computing $P(Y_i|Y_{i-1}, \ldots, Y_0, \mathbf{C})$ iteratively. Due to their extensive training on a massive amount of data, LLMs are good estimators of $P(Y_0|\mathbf{C})$ (Brown et al., 2020). However, LLMs are also prone to hallucinations when providing long answers: they deviate from the instructions or state false information (Huang et al., 2023). Indeed, estimating the true conditional distribution of the full output $P(\mathbf{Y}|\mathbf{C})$ is more challenging, especially when $\mathbf{Y}$ is long as it requires building a probability tree considering all possible values for the intermediate $Y_i$. This tree can be approximated using beam search or heuristic-guided tree search algorithms (Yao et al., 2023; Wan et al., 2024). However, their performance is still dependent on the output length. We alleviate this problem by conditioning the inference query on the causal parents of the output, i.e. instead of providing the complete context $\mathbf{C}$ as an input of the LLM, we use the much smaller subset $\mathbf{pa}(\mathbf{Y}) \subset \mathbf{C}$. Assuming knowledge of the causal graph, this choice can greatly reduce the size of the context window and mitigate hallucination. In the rest of this section, we assume that the LLM can provide a close estimate of the true conditional distribution $P(\mathbf{Y}|\mathbf{pa}(\mathbf{Y}))$. We challenge this assumption in our experiments. We investigate LLMs' causal inference abilities in counterfactual settings given the estimated causal model $\mathcal{M}$. Counterfactual queries answer the question: "How would variable $Y$ change if we had $X = x$ instead of $X = x'$?". This question can be answered by performing *abduction*, *intervention* and *prediction*. The abduction step estimates the values of the exogenous factors $U$ from the observed quantities: $P(U|x', y')$. The intervention step edits the

causal graph with the $do(X = x)$ operation. The prediction step computes the remaining variables from their parent values: $P(Y|\mathbf{pa}(Y))$. The corresponding quantity is expressed as follows:

$$P(Y|do(x), x', y') = \sum_{u \in U} P(Y|do(x), u)P(u|x', y') \tag{1}$$

Figure 2 illustrates these steps. To be efficient, we approximate some of them. We sample a single $u \sim P(U|\mathbf{ch}(U))$ using the LLM. $\mathbf{ch}(\cdot)$ represents the children of $U$. We perform the abduction and prediction steps only on the variables affected by the intervention, as shown in Figure 2e where we use the LLM as described above to compute $P(A|X, B, U)$ and $P(Y|A)$.

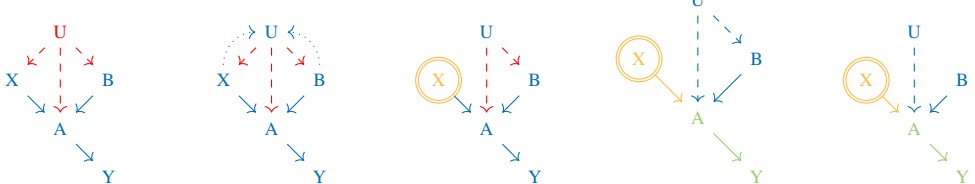

(a) Factual graph.    (b) Abduction step.    (c) Intervention step.    (d) Prediction step.    (e) Optimised version.

Figure 2: Counterfactual inference steps. (2a) The original causal graph. (2b) We estimate the possible values of the exogenous factors, here $U$, from the observations. (2c) We perform the $do(X = x)$ operation. (2d) We predict the values of the remaining variables given their parent causes. Here, $X$ and $U$ are known. $B$, $A$ and $Y$ should be predicted. (2e) However, to maintain efficiency, we consider a single possible value per exogenous factor and re-compute only the variables affected by the intervention: here $B$ is unaffected and not re-computed.

## 4 INFERENCE ON SYNTHETIC DATA

### 4.1 EXPERIMENTAL SETUP

We verify the applicability of our method on synthetic data and use it to investigate the current limitations of LLMs on counterfactual reasoning tasks. Cladder is a synthetic dataset containing small self-contained causal graphs of three of four variables with no unobserved confounders (Jin et al., 2023). Queries in Cladder test the causal capabilities of a model. We focus on the counterfactual subset. We extract the results for the counterfactuals queries (rung 3, det-counterfactual)[1]. Queries are divided into commonsense, nonsensical and anti-commonsense categories. Nonsensical queries are composed of abstract variables not conveying any semantic meaning. Anti-commonsense queries contain common concepts as variables but with fictive causal relationships. Figure 3 shows an example of anti-commonsense query from Cladder. We conduct experiments using LLaMA-3.1 (Dubey et al., 2024), GPT-3.5 (Ouyang et al., 2022), GPT-4 (version 1106), GPT-4o and GPT-4o-mini (OpenAI, 2023; 2024). We query GPT models via the OpenAI API while we run LLaMA-3.1 locally on one GPU NVIDIA A100 using Ollama. We use Langchain to interface with the LLMs. We use the default hyperparameters of both models and allow 12 refinement steps to format the LLM answers properly. When the answer does not match the expected format, we add a parsing layer. The prompts used are given in Appendix B. Our framework is denoted *Counterfactual-CI*. We compare our method with baseline LLM models from (Jin et al., 2023).

### 4.2 EVALUATION RESULTS

We compare the results using our framework against basic and causal prompting (Jin et al., 2023). To provide insights into the abilities and limitations of LLMs, we also create ablated models. These models, denoted with $\mathcal{G}_{gt}$ are provided with the ground-truth graph (extracted by parsing the input query) and are only tasked to perform the counterfactual inference step. Table 1 describes the obtained

---

[1]We download the results divided by query type and commonsense here: `https://edmond.mpg.de/dataset.xhtml?persistentId=doi%3A10.17617%2F3.NVRRA9`.

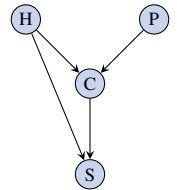

(b) Ground-truth graph.

> **Example of Anti-Commonsense Counterfactual Query**
>
> Imagine a self-contained, hypothetical world with only the following conditions, and without any unmentioned factors or causal relationships: Unobserved confounders has a direct effect on drinking coffee and salary. Proximity to a college has a direct effect on drinking coffee. Drinking coffee has a direct effect on salary. Unobserved confounders is unobserved. We know that confounder active or close to a college causes drinking coffee. confounder active or drinking coffee causes high salary. We observed the person lives close to a college and confounder inactive.
>
> - - - - - - - - - - - - - - - - - - - - - - - - - -
>
> Would the employee has a high salary if drinking coffee instead of not drinking coffee?

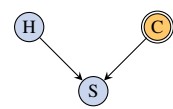

(c) Counterfactual graph.

(a) Prompt in natural language. The correct answer is 'yes'.

Figure 3: Example of counterfactual query from the Cladder dataset. (left) The context and question description in natural language as provided to the model. (right) The corresponding ground-truth and counterfactual causal graph with (H) a hidden confounder, unlike in real-world situations, its value is given in the dataset and thus shown in blue, (C) coffee drinking, and (S) a high or low salary. All causes affecting the system are mentioned. Intervention is shown in yellow.

results. We do not include parsing errors in the computation of the results as we aim to study the abilities of the LLMs on causal inference tasks separately from their capacity to follow instructions and generate structured outputs. This is not an issue for most models as they only contain a small number of uniformly distributed errors (see Figure 4). Most LLMs do not perform significantly better than random guessing (50% accuracy). There is no consistent difference of accuracy between the three levels of commonsense, showing little bias towards prior knowledge. The best models are GPT-4-1106+Causal CoT and Counterfactual-CI-GPT-4o-$\mathcal{G}_{gt}$ although their performance remains limited (around 10% better than random guessing). Our framework decomposes counterfactual reasoning into a sequence of atomic steps, allowing us to get insights into the LLMs' causal reasoning abilities. We observe an improvement of performance when the causal graph is given to the LLM. This is most noticeable with GPT-4o. It indicates that LLMs are not systematically able to recover the true causal structure even when no information is hidden. However, the improvements are often small, e.g. no improvement is observed for GPT-3.5, highlighting that the accuracy seldom depends on the access to the correct causal structure. Our framework ensures that the right causal factors are provided to the appropriate causal variables, i.e. the value of a variable is solely determined by the value of its parents or from an intervention. Therefore, the bottleneck in accuracy lies in the computation of the functions $P(\mathbf{Y}|\mathbf{pa}(\mathbf{Y}))$. We provide an example of failure case illustrating this limitation in LLMs in Section 4.4.

We look deeper into Figure 4 and Table 2. Figure 4 shows the decomposition of the results between graph building and inference errors. Table 2 further shows the Graph Edit Distances (GED) between the causal graphs built by the models and the ground-truth causal graphs. The GED counts the number of node and edge edits required to transform the first graph to the second. We can see in Figure 4 that most LLMs can accurately build a causal graph. Only LLaMA-3.1 shows a high number of errors during the causal graph generation. Moreover, Table 2 shows that GPT-4o, GPT-4o-mini and GPT-3.5 require less than one modification in average to recover the true graph (GED metric). As the GED$_{topology}$ metric is very close to the GED, it indicates that a semantic difference is systematically associated with a structural difference. This is further confirmed by the observed difference between the GED$_{topology}$ and IoU-GED$_{topology}$ metrics.

## 4.3 LLM SELF-EVALUATION

We ask the LLM to self-evaluate its generated factual and counterfactual graphs. Each sample in the dataset contains a context document a query. Thus, one factual graph corresponding to the context and a second counterfactual graph with the intervention are generated for each sample. We summarise

Table 1: Accuracy on the counterfactual subset of the Cladder dataset. Only extracted answers are shown. Accuracy is reported overall and divided by commonsense (Common.), nonsensical and anticommonsense (Anti-Common) queries. Models with $\mathcal{G}_{gt}$ are given the true causal graph extracted via standard parsing. Results with * are obtained on $\sim 65\%$ of the dataset and cannot be directly compared with the other models (see Figure 4). LLMs do not demonstrate good counterfactual inference abilities even when the causal and reasoning structures are given, highlighting that the performance bottleneck lies in the LLMs' ability to perform accuracte prediction.

| | Accuracy | | | |
| --- | --- | --- | --- | --- |
| | Overall | Common. | Nonsensical | Anti-Common. |
| LLaMA | 56.61 | 54.99 | 58.26 | 54.78 |
| GPT-3 Non-Instr. (davinci) | 50.00 | 47.01 | 49.17 | 47.78 |
| GPT-3 Instr. (text-davinci-001) | 50.07 | 53.28 | 50.82 | 45.22 |
| GPT-3 Instr. (text-davinci-002) | 51.76 | 54.13 | 51.93 | 48.99 |
| GPT-3 Instr. (text-davinci-003) | 58.02 | 54.13 | 59.23 | 59.42 |
| GPT-3.5 | 50.49 | 51.85 | 51.38 | 47.25 |
| GPT-4-1106 | 59.77 | *61.25* | 59.78 | 58.26 |
| GPT-4-1106 + CausalCoT | **62.31** | **63.53** | 60.06 | **65.78** |
| Counterfactual-CI-GPT-4-1106* | 50.57 | 51.17 | 49.33 | 52.94 |
| -GPT-4o | 52.26 | 53.85 | 51.39 | 52.37 |
| -GPT-4o-mini | 51.86 | 54.19 | 51.22 | 50.80 |
| -GPT-3.5 | 52.31 | 48.39 | 53.57 | 53.92 |
| -LLaMA-3.1* | 52.11 | 53.00 | 53.11 | 48.39 |
| -GPT-4o-$\mathcal{G}_{gt}$ | *60.53* | 58.68 | **61.23** | *61.20* |
| -GPT-4o-mini-$\mathcal{G}_{gt}$ | 56.58 | 53.16 | 58.03 | 56.91 |
| -GPT-3.5-$\mathcal{G}_{gt}$ | 49.80 | 47.78 | 48.41 | 54.52 |
| -LLaMA-3.1-$\mathcal{G}_{gt}$ | 58.05 | 54.33 | *61.17* | 54.79 |

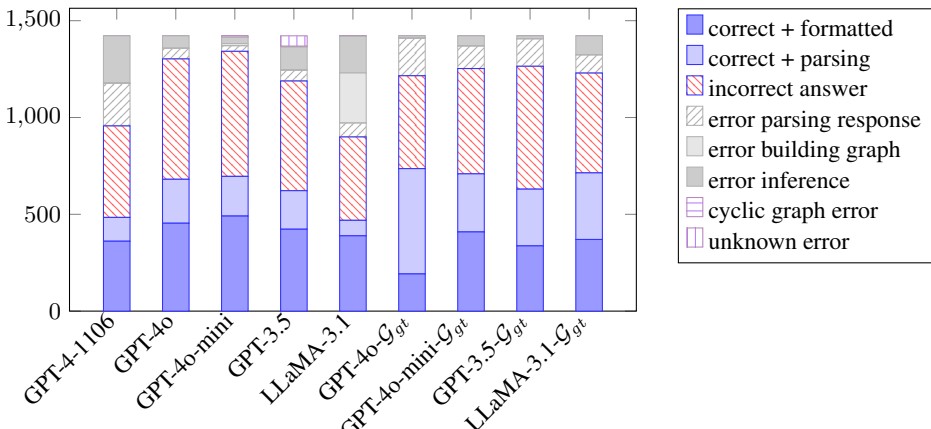

Figure 4: Partition of the Counterfactual-CI models results between correct, incorrect answers and errors. Errors in grey are not considered as counterfactual reasoning errors but as instruction errors and are not considered in the results of Table 1. Models can usually generate the causal structure and conduct inference. GPT-4-1106 and LLaMA-3.1 show a lower capacity to follow instructions and generate structured outputs. Models also often require to have their response parsed to extract the answer, particularly GPT-4o-$\mathcal{G}_{gt}$.

each graph into text format and prompt the LLM to return a plausibility score for the chain of events and a confidence score for its prediction (prompts are given in Appendix B.4). Table 3 shows the average LLM self-evaluation on the accuracy of the generated causal graphs. The models expectedly attribute a slightly lower plausibility to the counterfactual graphs. Models attributing a higher score also show a higher confidence. In addition, the models that obtain a higher accuracy in Table 1 tend to attribute lower scores and confidence than their counterparts. We put these results in perspective with the evaluation on real-world graphs in Section 5.

Table 2: Graph distances with ground-truth graph. GED stands for Graph Edit Distance. IoU-GED is the GED metric between the intersection and the union of the built and ground-truth graphs. The base metric matches the variable names while the topology metric only look at the structure. All graphs have either three and four nodes. Most models require less than one change in average, except for GPT-4o-mini and LLaMA-3.1.

| | GED | IoU-DEG | $GED_{topology}$ | $IoU\text{-}GED_{topology}$ |
|---|---|---|---|---|
| Counterfactual-CI-GPT-4-1106 | 0.814 | 3.929 | 0.814 | 2.240 |
| -GPT-4o | 0.897 | 4.268 | 0.897 | 2.100 |
| -GPT-4o-mini | 2.667 | 5.898 | 2.666 | 6.175 |
| -GPT-3.5 | 0.582 | 3.708 | 0.579 | 0.919 |
| -LLaMA-3.1 | 2.420 | 6.198 | 2.419 | 4.790 |

Table 3: Self-evaluation and confidence provided by the LLMs. Results for GPT-3.5 are omitted because the model did not return properly formatted scores on a sufficient number of samples (less than five). Models attribute a slightly lower score to the counterfactual graphs. Models performing better tend to attribute lower scores and confidence than their counterparts.

| | Self-Evaluation | | Self-Confidence | |
|---|---|---|---|---|
| | Factual | Counterfactual | Factual | Counterfactual |
| GPT-4-1106 | 0.689 | 0.650 | 0.899 | 0.907 |
| GPT-4o | 0.506 | 0.489 | 0.666 | 0.707 |
| GPT-4o-mini | 0.501 | 0.452 | 0.732 | 0.722 |
| GPT-3.5 | - | - | - | - |
| LLaMA-3.1 | 0.772 | 0.764 | 0.820 | 0.829 |

## 4.4 EXAMPLE OF REASONING FAILURES

We show an example of failure case with GPT-4o for the example provided in Figure 3. The model explanation is given below:

```
The target variable 'salary' is influenced by two parent causes: 'drinking coffee' and '
    unobserved confounders'. Given that the person drinks coffee (true), we might expect the
    salary to be positively affected. However, the status of unobserved confounders is
    inactive, which suggests a lack of additional income influence. Thus, the overall effect
    results in a low salary.
```

Although the model generates the correct causal factual and counterfactual graphs (as shown in the figure), the inference step fails. The model answers 'low salary' instead of 'high salary'. The context specifies that 'confounder active or drinking coffee causes high salary' but the LLM makes a mistake and interprets it as a logical AND instead of a logical OR. This example illustrates the type of prediction error that is prevalent in the LLMs' reasoning. It can be compared with the LLM limitations in robust and abstract reasoning tasks (Wu et al., 2023; Gendron et al., 2024; Jin et al., 2024).

## 5 EXPERIMENTS ON REAL-WORLD USECASE

### 5.1 EXPERIMENTAL SETUP

We extract 5,486 media events related to the "Price of Oil" from EventRegistry (Leban et al., 2014). These events, spanning the first quarters of 2015, 2020, 2022, and 2023, were selected based on geopolitical events highlighted by the U.S. Energy Information Administration and the Russo-Ukrainian War [2]. We perform experiments using LLaMA-3.1 (Dubey et al., 2024) and GPT-4o (OpenAI, 2023; 2024). We provide early results on real-world news documents. Due to the unavailability of the ground truth, we focus our experiments on a handful of documents that we manually verify.

---

[2]The geopolitical events were highlighted in the following report, last accessed on August 25[th] 2023: https://www.eia.gov/finance/markets/crudeoil/spot_prices.php.

## 5.2 Counterfactual Inference

Table 4: Example of counterfactual inference performed by the Counterfactual-CI model on real-world data. The first row shows the extracted factual graph and the counterfactual graph under interventions (in yellow) *do(0='low')* and *do(9='False')*. The exogenous factor is in red. After abduction, a value is assigned to it (illustrated in blue). Variables inferred during the prediction step are shown in green. The bottom rows show the values of the factual and counterfactual worlds.

| | Factual Graph | Intervened Graph |
| --- | --- | --- |

| | Factual Values | Counterfactual Values | |
| --- | --- | --- | --- |
| | | GPT-4o | LLaMA-3.1 |
| 0 | Severity of COVID-19 pandemic (range element): severe | low (from do operation) | |
| 1 | Severity of oil price war (range element): severe | | |
| 2 | Bursa Malaysia downtrend magnitude (int): 29% | 20 | high |
| 3 | FBM KLCI index value (float): 1,280.63 | 1580 | 1500.00 |
| 4 | Selling pressure on stocks (range element): high | | |
| 5 | Investors moving into cash (bool): True | | |
| 9 | Malaysia's change of coalition government (bool): True | False (from do operation) | |
| 10 | Downside risks to corporate earnings (range element): high | low | low |
| 11 | Travel restrictions imposed worldwide (range element): severe | none | none |
| 12 | Condition of oil & gas and airlines sectors (range element): bad | good | good |
| h0 | Potential end of COVID-19 pandemic (bool): None | False | True |

Table 4 provides an example of results obtained with our method for a model $\mathcal{M}(D)$ extracted from a document $D$ and under interventions $\mathcal{M}_{0='low',9='False'}(D)$. The input text and the explanations given by the LLMs during inference are given in Appendix C. The document $D$ describes how the COVID-19 pandemic and the rise in oil prices affect Malaysia's economy. We perform two interventions on the graph and build a counterfactual world where the severity of the pandemic is low and a change of governement has not happened. From these interventions, the model updates the rest of the variables and predicts a better economical situation. We emphasise that these results should be taken as an illustrative example only. We also observe that the two models return different values for the exogenous factor but reach the same conclusions.

As in Section 4.3, we ask the LLM to self-evaluate its generated graphs. We generate four causal graphs from a single document and, for each graph, we automatically build six counterfactual queries. We summarise each graph into text format and prompt the LLM to return a plausibility score for the chain of events and a confidence score for its prediction (prompts are given in Appendix B.4). We show the results in Table 5. The LLMs generally provide higher scores to the factual graph than to the counterfactuals. This is not surprising as the former are extracted from real data, closer to the LLMs' training distribution. However, they still give high scores to the built counterfactuals. GPT-4o is also more consistent on the factual graphs, with a lower standard deviation, highlighting consistency between graph generations. LLaMA-3.1 shows a high standard deviation for all results.

GPT-4o gives higher scores and confidence than on the synthetic data despite the more complex causal structure. However, LLaMA-3.1 provides lower scores, particularly for the counterfactual graph. We hypothesise that lower scores are attributed to the counterfactual graphs because they break the causal reasoning chains via the intervention. It hints that the LLMs rely on commonsense clues and already observed reasoning chains from their training distribution to build their answers. This can be further observed in the explanations provided in Appendix B.4 and was described in the context of abstract reasoning by (Gendron et al., 2024).

Table 5: Self-evaluation and confidence provided by the LLMs on the plausibility of the described set of events and their causal relationships for the document described in Appendix B. The average of three end-to-end runs is shown.

| | Self-Evaluation | | Self-Confidence | |
|---|---|---|---|---|
| | Factual | Counterfactuals | Factual | Counterfactuals |
| Counterfactual-CI-GPT-4o | $0.850 \pm 0.000$ | $0.739 \pm 0.129$ | $0.863 \pm 0.022$ | $0.762 \pm 0.155$ |
| -LLaMA-3.1 | $0.563 \pm 0.330$ | $0.256 \pm 0.340$ | $0.600 \pm 0.346$ | $0.367 \pm 0.400$ |

## 6 LIMITATIONS

LLMs provide free-text answers that can be different from the format provided in the instructions. This poses challenges for building an end-to-end pipeline that requires using LLMs at multiple stages, as the responses can be difficult to parse automatically, and errors can accumulate. In our future work, we will include fine-tuning in our pipeline to mitigate this issue. Due to the high cost of running LLMs or accessing them through APIs, we only tested our method on synthetic data and short text snippets. We intend to apply it to larger amounts of data in the future. We also only consider DAG structures, whereas real-world events can contain feedback loops. We will integrate them into our future work. As the LLM discovers the causal structure, errors can be present, and confounders can be omitted. The counterfactual results depend on the causal graph to be accurate. In addition, our current approach for real-world data proxies ground-truth counterfactual data by retrieving such values from LLMs. Our future work will focus on verifying the accuracy of the intermediate steps and building ground-truth counterfactual data.

## 7 CONCLUSION AND BROADER IMPACT

We propose an end-to-end method for conducting causal structure discovery and counterfactual causal inference from unstructured natural language. We demonstrate the applicability of our method on real-world news events, showcasing the LLMs' abilities to perform causal discovery and inference not as a standalone model but as a part of a larger framework. Furthermore, our experiments show that LLMs can extract the causal structure of a piece of text but fail during the reasoning part. This expands previous findings on the limitations of LLMs on reasoning tasks (Wu et al., 2023; Gendron et al., 2024; Jin et al., 2024).

This research is still in its infancy but has shown promising results for computing causal inference and leveraging LLMs. As the inference process is divided into several independent computations of causal variables given their parents, this configuration provides inherent interpretability and allows auditing an LLM's answer. Our future work will follow two lines of research. On one hand, as LLMs have been shown to perform better by using refinement techniques (Wu et al., 2023; Madaan et al., 2023; Yuan et al., 2024; Qiu et al., 2024), our counterfactual self-evaluation method could be used to conduct *Counterfactual Self-Learning*: refine LLMs' answers to improve them and teach LLMs to reason more causally by providing them with counterfactual data (Bareinboim et al., 2022). On the other hand, we aim to create a framework to build and test counterfactuals based on real values and validate whether LLM-extracted causal relationships hold. Doing so would avoid LLM hallucinations and counterfactual reasoning deficiencies similar to the ones shown in Section 4 and lead to robust causal reasoning, where every piece of information could be traced back to evidence from the real world. We expect that such a framework will have extensive applications in many domains, extending the use of causal reasoning to any domain where text is available. In particular, we expect that could lead to a greater automation of strategic foresight, democratizing and enabling a wider use of it to enhance decision-making at all societal levels.

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

## A    GRAPH SUMMARISATION

When considering several documents, we build their corresponding causal graphs independently and then attempt to merge them. The problem of combining multiple causal graphs together is known as the *Structural Causal Marginal problem* (Gresele et al., 2022). This is a challenging problem that we approach from two different angles. Considering two causal models $\mathcal{M}_1$ and $\mathcal{M}_2$, we first generate embeddings for every node of the two graphs by extracting their representation generated by the last hidden layer of the LLM. The prompt of a node is a concatenation of all its attributes. We omit its current value because we want to generate a representation of the variable and not the current instance. Motivated by the success of Graph Neural Networks (GNNs) at aggregating neighborhood information in graphs (Kipf & Welling, 2017; Velickovic et al., 2018; Xu et al., 2019), we also add to he prompt the attributes of its immediate neighbours and how they are connected via the edge attributes. Then, we use a clustering algorithm to find nodes that share similar latent representations. We select DBSCAN (Ester et al., 1996). Since it is a density-based algorithm, it allows use to restrict the sparsity of the clusters and cluster together nodes with only very close representations.

After extracting similar nodes, we consider two ways to combine graphs: *summarisation* and *analogy*. Summarisation implies considering similar nodes as a single node in the merged graph, inheriting the edges of all the nodes in the set. This approach is straightforward and allows reducing the number of nodes as the graph grows. However, assessing if two causal variables can be merged is a challenging problem. In particular, in the absence of a lot of observations (one text only shows one observation per variable), the merged causal graph can be easily falsified (Gresele et al., 2022).

Analogy merging is inspired by the research on analogical reasoning (Osta-Vélez & Gärdenfors, 2022; Forbus, 2015). We view similarity between nodes representations as an indication that analog mechanisms are causing them. To represent an analogy, we do not modify the existing graphs but add a common unobserved ancestor between similar nodes to integrate the similarity information from the clustering process without making assumptions regarding the nature of their similarity. The merged graphs can share information via backdoor paths. Unlike summarisation, analogy does not remove nodes but adds more. However, this method does not introduce ways to falsify the graphs and preserve the mechanisms of the initial graphs. The two approaches are illustrated in Figure 5.

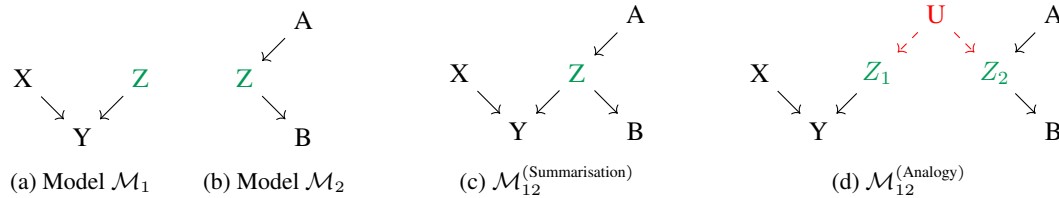

(a) Model $\mathcal{M}_1$      (b) Model $\mathcal{M}_2$      (c) $\mathcal{M}_{12}^{(\text{Summarisation})}$      (d) $\mathcal{M}_{12}^{(\text{Analogy})}$

Figure 5: Illustration of graph merging process for two models $\mathcal{M}_1$ and $\mathcal{M}_2$. We assume that the common variable $Z$ is the same in the two graphs and should be combined. Figure 5c shows the merged graph using summarisation: $Z$ is shared by the two mechanisms. Figure 5d shows the merged graph using analogy: the variables $Z_1$ and $Z_2$ from $\mathcal{M}_1$ and $\mathcal{M}_2$ remain separated but share a common ancestor.

## B    PROMPTS USED

### B.1    CAUSAL STRUCTURE DISCOVERY

Here is the system prompt used for causal discovery:

```
Your task is to summarise a text into a JSON dictionary of instantiated causal variables and
    the causal relationships between them.
Variables should be as atomic and detailed as possible. Causal relationships should describe
    how the value of the first variable affects the value of the second.
One sentence usually describes two or more variables and connects them. For each variable, the
    following questions should be answered:
'What are the causes of this variable's value? Is it fully explained by the available
    information or are some causes missing?'
If some causes seem to be missing, create new (hidden) variables.
```

```
Hidden variables represent missing information to fully explain the value of one or more
    observed variables.
They cannot have incoming edges. Identify the major and minor variables and how they are
    connected.
Add the missing unknown variables when necessary. Follow carefully the instructions and write
    down your answer using only the given JSON format very strictly.
The format is as follows:
{
  "observed_nodes": [
    {
      "node_id": (str) "0",
      "description": (str) "<high-level short atomic description of causal variable 0>",
      "type": (str) "<variable type: e.g. bool, int, set element, range element>",
      "values": (str) "<set of possible values, if applicable>",
      "current_value": (str) "<current value>",
      "context": (str) "<contextual information type> : <value of the contextual information
          linked to the current instance>"
    },
    ...
  ],
  "hidden_nodes": [
    {
      "node_id": (str) "h0",
      "description": (str) "<high-level short atomic description of the hidden causal variable
          >",
      "type": (str) "<variable type: e.g. bool, int, set element, range element>",
      "values": (str) "<set of possible values, if applicable>",
      "current_value": (str) "", # This field is left empty because the current value of the
          variable is unknown since the variable is hidden
      "context": (str) "<contextual information type> : <value of the contextual information
          linked to the current instance>"
    },
    ...
  ],
  "observed_edges": [
    {
      "source_node_id": (str) "0",
      "target_node_id": (str) "1",
      "description": (str) "<high-level short atomic description of the causal relationship from
          variable 0 to 1>",
      "details": (str) "<detailed explanation of how the value of variable 0 affects the value
          of variable 1 in the text>"
    },
    ...
  ],
  "hidden_edges": [
    {
      "source_node_id": (str) "h0",
      "target_node_id": (str) "1",
      "description": (str) "<high-level short atomic description of the causal relationship from
          hidden variable 0 to 1>",
      "details": (str) "<detailed explanation of how the value of hidden variable 0 affects the
          value of variable 1 in the text>"
    },
    ...
  ]
}
```

Here is the parameterised user prompt, specific for each instance. The {text} parameter is replaced by the input document.

```
Here is the input text:
```
{text}
```
```

## B.2 GRAPH SUMMARISATION

When conducting graph summarisation, we do not use the LLM as a generative model but as an embedding model. We only provide node information using the following format. Texts in curly brackets represent variable parameters.

```
description: {description}, type: {type}, values: {values}, context: {context}
```

When adding neighbour information, we concatenate the initial representation with the folowing neighbour representation:

```
neighbour at distance {rank} from node: description: {description}, type: {type}, values: {
    values}, context: {context}
```

## B.3   COUNTERFACTUAL INFERENCE

During inference, we provide parent variables to the LLM and prompt it to estimate the value of the target variable. Here is the system prompt:

```
Your task is to predict the value of the target variable given its description, type, possible
    values, and context, and the attributes and values of its parent causes and the
    relationships connecting them.
The value of the target variable is fully determined by its direct list of causes. Reason step-
    by-step. Start by describing the attributes of the target variable and explain in your
    own words its relationships with its parent causes, how the variables are linked, and how
     their values cause the value of the target. Then, predict the value of the target
    variable. Provide a confidence score as a float between 0 and 1. Follow strictly the
    provided format.
```

The user prompt provides information about the target variable. The format is as follows:

```
The target variable has the following attributes: {node attributes}.
It is caused by the following variables:
```

The list of parent is then provided. The $i$th parent variable is described as follows:

```
{i}. {parent attributes}. Its value is {parent value}. Its causal relationship with the target
     is described as follows: {edge attributes}
```

We generate the value of the intervened variable using an LLM. Here is the system prompt provided for this task:

```
Your task is to interpret the attributes of a variable and propose an alternative/
    counterfactual instantiation different from its current value. The variable is described
    by its description, type, possible values, current value, and context. The counterfactual
     value should be a plausible alternative instantiation of the variable given the context,
     type, description, and possible values. Reason step-by-step. Start by describing the
    attributes of the variable and explain in your own words the reasons for the choice of
    the counterfactual value. Then, state the factual value and propose the new
    counterfactual value. Provide a confidence score as a float between 0 and 1. Follow
    strictly the provided format.
```

Here is the user prompt:

```
The variable has the following attributes: description: {description}, type: {type}, possible
    values: {values}, context: {context}. The current value is {current_value}. Propose a
    counterfactual value.
```

## B.4   EVALUATION

The evaluation of the causal factual and counterfactual models is performed by an LLM using the following prompts. Here is the system prompt:

```
Your task is to evaluate the plausibility of a set of events linked by causal relationships.
    The events are described by a high-level description and a value. The events are linked
    by causal relationships. The causal relationships are described by a high-level
    description. The overall plausibility of the set of events corresponds to the
    factorization of the plausibility of each event's occurrence given its causes. Reason
    step-by-step. Start by describing the events and the causal relationships. Explain in
    your own words the reasons for the plausibility of each event. Finally, provide an
    overall score for the plausibility of the sequence of events. Give an explanation
    describing your reasoning. Provide an overall confidence score as a float between 0 and 1.
     Follow strictly the provided format.
```

The user prompt describes the events in the topological order of the causal graph. Before each event, the causal relationships with its parents is also described. Here is an example:

```
The causal graph is composed of the following events:
({parent rank} -> {target rank}) {edge description}.
{target rank}. {target description}. The value is {node current_value}
```

## C  SUPPLEMENT TO THE EXPERIMENTS

In this section, we describe in more details the counterfactual causal inference example from Table 4. The document from which is extracted the causal graph is shown below:

```
KUALA LUMPUR: Bursa Malaysia downtrend could be far from over as there is always more room to
    decline depending on the severity of COVID-19 pandemic and oil price war, said economists
    after key benchmark FBM KLCI took another beating in the early trading session yesterday.

To what may be seen as continued selling pressure from last week's Friday the 13th, the FTSE
    Bursa Malaysia KLCI (FBM KLCI) lost 44.99 points to 1,299.76 at 9.10am yesterday,
    compared with Friday's close of 1,344.75, after opening 25.38 points lower at 1,319.37
    yesterday morning.
At market closing yesterday, the FBM KLCI closed at 4.77 per cent lower to 1,280.63 points
    with turnover at 4.473 billion shares valued at RM3.687 billion.
Bank Islam chief economist Dr Mohd Afzanizam Abdul Rashid said if history is of any guide, the
    FBM KLCI has fallen sharply between January 11, 2008 (1,516.22 points) and October 29,
    2009 (829.41 points).
He said during the Asia Financial Crisis in 1997/1998, the FBM KLCI was down massively by 79.2
    per cent between February 25, 1997 (1,265.01 points) and September 1, 1998 (262.7 points)
    .
"For now, the FBM KLCI touches its peak at 1,895.18 points on April 19, 2018 and has plunged
    by 29.0 per cent to 1,344.75 points as of March 13, 2020. There is always more room to
    decline obviously due to the virus outbreak and oil price war," he told NST Business.
CGS-CIMB analyst Ivy Ng Lee Fang said the research firm has cut its year-end FBM KLCI target
    to 1,449 points.
"We advise investors to seek shelter in defensive and high-dividend-yield stocks until the
    concerns over the global spread of COVID-19 subside.
"These, together with Malaysia's unexpected change of coalition government, could pose
    downside risks to corporate earnings, which are difficult to measure at this time," it
    said.
Ng said during the global financial crisis, FBM KLCI fell by 45 per cent from its peak to 829
    points, its lowest ever and FBM KLCI earnings fell by 8.7 per cent in 2009.
She said there is a potential earning downside risk of 10.3 per cent to its current 1.6 per
    cent FBM KLCI earnings per share growth forecast if the earnings risk resembles that of
    during the global financial crisis decline (-8.7 per cent).
Meanwhile, AxiCorp market strategist Stephen Innes said small and medium enterprises (SMEs)
    are most at risk given they generally operate on small operating cushions and will need
    to dive into credit lines if not hefty interest credit card to keep things afloat.
"Unless governments actually get money physically into the hands that need it the most things
    can turn bad. This will put pressure on bank stocks due to credit line drawdown and an
    increase in defaults.
"We know oil and gas and airlines are in a bad way due to travel restrictions imposed around
    the world.
"At this stage, I see no way out and expect more investors to move into cash or keep
    investment portfolio very light primarily in recessionary proof areas like healthcare.
"But honestly with credit under duress, I think cash is the place to be. The only endgame in
    sight is when the world sees COVID-19 ends," he said.
```

Table 6 provides the explanations given by the models when inferring the values of the remaining variables during the prediction step. We can observe that the two models reason similarly, although GPT-4o tends to return a more precise and detailed explanation. The only difference in behaviour is for variable $h0$ where GPT-4o does not answer. As this value is computed during the abduction step, in an anticausal fashion (predict a cause from its consequences), this is expected. This issue can be alleviated by marginalising over the set of possible values of $h0$.

Tables 7 and 8 provide examples of explanation returned by the models when evaluating the factual and counterfactual causal graphs. GPT-4o provides longer and more detailed explanations while the justification provided by LLaMA-3.1 is not always very explanative. We can see that scores and confidence are highly correlated for both models although outliers can also be observed.

Table 6: Explanation of the inferred values by GPT-4o and LLaMA-3.1 for every variable in Table 4.

|  | GPT-4o | LLaMA-3.1 |
|---|---|---|
| 2 | The Bursa Malaysia downtrend magnitude is influenced by several factors. The severity of the COVID-19 pandemic is low, which would typically suggest a smaller downtrend magnitude. However, the severity of the oil price war is severe, which greatly increases the downtrend magnitude. The downside risks to corporate earnings are low, which would also typically suggest a smaller downtrend. Lastly, the potential end of the COVID-19 pandemic is not expected (False), which does not help reduce the downtrend magnitude. Thus, considering the severe impact of the oil price war, I predict a moderate downtrend magnitude of around 20%. | Given that Severity of oil price war is severe, increasing Bursa Malaysia downtrend magnitude and Downside risks to corporate earnings are low, contributing less to the downtrend. Although Potential end of COVID-19 pandemic would reduce the downtrend magnitude, Severity of COVID-19 pandemic being low would increase it, but this effect is expected to be lower than that of oil price war. Hence Bursa Malaysia downtrend magnitude is predicted to be high. |
| 3 | The FBM KLCI index value is influenced by several factors for the market close yesterday. The Bursa Malaysia downtrend magnitude shows a significant percentage decline from peak (20%), which has a negative impact on the index. Furthermore, the selling pressure on stocks is high, which also decreases the index value. On the other hand, the condition of the oil & gas and airlines sectors is good, which supports the index value. Given these mixed factors, the FBM KLCI index is estimated to be lower but not drastically so. | The estimated value of the FBM KLCI index is 1500.00 due to the high Bursa Malaysia downtrend magnitude and high selling pressure on stocks, which both decrease the FBM KLCI index value. However, the good condition of oil & gas and airlines sectors has a positive effect on the index value. |
| 10 | The downside risks to corporate earnings are influenced by Malaysia's change of coalition government. Since the change of coalition government did not occur (value is False), the downside risks are expected to be low. | *(no explanation returned by the model)* |
| 11 | Given that the severity of the COVID-19 pandemic is low and the potential end of the pandemic is false, there would be no severe travel restrictions imposed worldwide. The low severity of the pandemic suggests minimal disruption, and since there's no potential end of the pandemic yet, current measures would likely remain in place but not be stringent. | Given that the severity of COVID-19 pandemic is low and there is a potential end of COVID-19 pandemic, it can be inferred that travel restrictions imposed worldwide would likely be none. This is because a low severity of the pandemic suggests that the situation is under control, and a potential end to the pandemic would further reduce the need for travel restrictions. |
| 12 | Since the travel restrictions imposed worldwide were 'none' yesterday, it is likely that the oil & gas and airlines sectors condition was 'good'. Severe travel restrictions would worsen these sectors, but in the absence of any restrictions, the condition should be favorable. | Given that yesterday there were no worldwide travel restrictions, it is estimated that the condition of oil & gas and airlines sectors was good. |
| h0 | The value cannot be determined as there are no parent causes provided to infer the outcome. | The potential end of COVID-19 pandemic will occur because all the parent causes have values that lead to this outcome. |

Table 7: Example of graph scores explanations given by GPT-4o in Table 5. FG stands for Factual Graph and CG for Counterfactual Graph. Sc stands for Score and Cf stands for confidence. We show the explanations for one graph and two counterfactual graphs.

|     | Sc   | Cf   | GPT-4o |
| --- | ---- | ---- | ------ |
| FG  | 0.85 | 0.85 | The sequence of events and their causal relationships are highly plausible given the context of the COVID-19 pandemic and global economic conditions. The COVID-19 pandemic had a significant impact on global markets, including Bursa Malaysia, leading to a downtrend in FBM KLCI points. The severity of the oil price war also contributed to this downtrend. Travel restrictions due to the pandemic affected both the oil and gas sector and the airline sector, leading to further economic pressure. The change in Malaysia's coalition government affected corporate earnings, contributing to the overall economic situation. The lack of financial support from the government increased pressure on bank stocks, especially given the bad operating conditions for SMEs. The values provided for FBM KLCI points and other financial metrics are consistent with the described events and their causal relationships. Therefore, the overall sequence of events is highly plausible. |
| CG1 | 0.5  | 0.5  | Let's break down the events and their causal relationships one by one: 1. Travel restrictions (Event 0) are false, meaning there are no travel restrictions in place. 2. The severity of the oil price war (Event 1) is high, which indicates a significant conflict impacting oil prices. 3. The severity of the COVID-19 pandemic (Event 2) is also high, suggesting widespread impact from the pandemic. 4. Travel restrictions affect the oil and gas sector condition (causal relationship 0 -> 3). 5. The condition of the oil and gas sector (Event 3) is bad. Given the provided events and relationships, the key causal link to evaluate is between travel restrictions (Event 0) and the oil and gas sector condition (Event 3). Since travel restrictions are not in place (false), this would typically suggest a neutral or positive impact on the oil and gas sector. However, the sector condition is bad. This could be due to other factors not explicitly mentioned in the graph, such as the high severity of the oil price war (Event 1) and the high severity of the COVID-19 pandemic (Event 2). The graph doesn't explicitly link these other factors to the oil and gas sector condition, but they are plausible contributors. Overall, the sequence of events seems somewhat plausible but lacks explicit causal links for all influencing factors, resulting in a moderate score and confidence. |
| CG2 | 0.3  | 0.5  | The causal graph has several issues that impact its overall plausibility. First, the event 'FBM KLCI earnings decline in 2009' is given a value of 5.0, which is plausible given the historical context of the 2008-2009 financial crisis. However, the subsequent events 'Severity of COVID-19 pandemic' and 'Severity of oil price war' are not assigned values, making it difficult to evaluate their plausibility. The link between 'FBM KLCI earnings decline in 2009' and 'Potential earning downside risk' is plausible, as past earnings declines can influence future downside risks. However, the lack of values for the severity of the pandemic and oil price war significantly reduces the confidence in the overall plausibility of the causal graph. |

Table 8: Example of graph scores explanations given by LLaMA3.1 in Table 5. FG stands for Factual Graph and CG for Counterfactual Graph. Sc stands for Score and Cf stands for confidence. We show the explanations for one graph and two counterfactual graphs.

|     | Sc  | Cf  | LLaMA-3.1 |
| --- | --- | --- | --- |
| FG  | 0.7 | 0.8 | The plausibility of the sequence of events is high due to the following reasons: COVID-19 pandemic (event 0) has a direct impact on recessionary trends (event 1). Fiscal policy responses (event 2) mitigated the effects of recessionary trends. However, oil price war (event 3) influenced FBM KLCI (event 4), which was also impacted by COVID-19 pandemic and recessionary trends. |
| CG1 | 0.6 | 0.7 | The event 'COVID-19 pandemic' with a value of True increases the likelihood of recessionary trends and has an impact on FBM KLCI. The event 'Oil price war' with a value of 80 influences FBM KLCI but is not directly linked to recessionary trends, despite its potential indirect impact. However, recessionary trends are False which decreases the likelihood of FBM KLCI being influenced by it. |
| CG2 | 0.0 | 0.8 | The COVID-19 pandemic has had a significant impact on the global economy, leading to recessionary trends. However, fiscal policy responses have mitigated some of these effects. The oil price war has also influenced the FBM KLCI, but its overall value remains high. |

