# OpenReview forum: "Counterfactual Causal Inference in Natural Language with Large Language Models"
_ICLR.cc/2025/Conference — Submitted to ICLR 2025_

### Official Review · Reviewer_Nzcs · 2024-10-21

**Soundness:** 3
**Presentation:** 3
**Contribution:** 2
**Rating:** 5
**Confidence:** 4

**Summary:**

They explore the challenges and potential of using large language models (LLMs) to infer causal relationships from unstructured text data, such as news articles. Then they present a method for discovering causal structures and conducting counterfactual causal inference using LLMs. They propose a pipeline approach:
1. They first use an LLM to extract the instantiated causal variables from text data and build a causal graph;
2.  Then they merge causal graphs from multiple data sources to represent the most exhaustive set of causes possible;
3. Finally, they conduct counterfactual inference on the estimated graph.

I think the paper is well-written. However, the main and only concern for me, for each of the aforementioned step above, they are many works already. To name a few:
+ Causal discovery with LLMs: (i) Efficient Causal Graph Discovery Using Large Language Models; (ii) Causal Graph Discovery with Retrieval-Augmented Generation based Large Language Models
+ Counterfactual inference with LLMs: (i) Using LLMs for Explaining Sets of Counterfactual Examples to Final Users; (ii) Do Models Explain Themselves? Counterfactual Simulatability of Natural Language Explanations


This work is more like a combination of different causal tasks with a novel end-to-end method. I admit there are merits for this end-to-end method but I think more discussion on the related work is necessary. I am open to other reviews' opinions about the novelty.

For this paper's soundness, I think this paper is well-written and the experiments are quite supportive.

**Strengths:**

1. The paper presents a novel approach by leveraging large language models to perform causal structure discovery and counterfactual inference from unstructured text.
2. The authors propose a comprehensive, end-to-end method that extracts causal graphs, performs interventions, and predicts counterfactual scenarios.
3. By incorporating causal graph conditioning, the method helps reduce biases inherent in LLM.

**Weaknesses:**

The main weakness is listed in their Limitation section, I agree with their pointed limitation and like their honesty:
1. The counterfactual setting is mainly on synthetic datasets.
2. The method currently focuses on DAGs and does not account for feedback loops, which are common in real-world events.

**Questions:**

1. I think the existing content(experiment&writting) is very sound. But the novelty is not that clear to me. The end-to-end way is more like a pipeline for different phases of causal tasks with LLMs. Namely, for both causal graph discovery and counterfactual inference, there are already existing works that use LLMs for this task. But I admit the end-to-end framework proposed in this paper is nove.
2. How do you validate the accuracy of the causal graphs generated by the LLMs, particularly when no ground-truth data is available for real-world scenarios? I guess this evaluation is based on synthetic datasets where the ground truth label is known?

---

> ### Author Response · Authors · 2024-11-22
>
> Thank you for the detailled review!
>
> Regarding your questions:
>
> 1. There indeed exists other studies on causal tasks with LLMs. Thank you for sharing some of them, we will include the missing ones in our related work. We would like to highlight the main differences with our work: (i) On causal discovery, most existing work focuses on recovering causal relationships that exist **implicitly** in the data (e.g. if a drug has an effect on a disease by reading patient data), this typically corresponds to tabular data, whereas in our work, we use LLMs to find causal relationships **explicitly** stated, from natural language sources of data. This is a different and less tackled problem. (ii) On counterfactual causal inference, we use a causal graph to divide the reasoning steps into atomic components (abduction, intervention and prediction). This is a different approach from existing work that allows us to study more deeply where the limitations of LLMs are when doing counterfactual causal inference and evaluate LLM predictions from causal parents only.
>
> 2. You are correct, we can only evaluate the ground-truth causal graphs with the synthetic data. In most real-world settings, we can only observe one factual world and do not have access to possible counterfactual ground truths (this is the fundamental problem of causal inference [1]). Due to this limitation, there are very few existing datasets that could be used for this task and the ones that exist may be solvable via commonsense reasoning and not true causal reasoning [2]. We will attempt to find or build datasets alleviating these shortcomings. We also look into LLM self-evaluation to study if LLMs can be good causal graph evaluators.
>
> [1] Judea Pearl. Causality. 2nd ed. Cambridge university press, 2009. doi: 10.1017/CBO9780511803161.
>
> [2] Zečević, M., Willig, M., Dhami, D. S., & Kersting, K. Causal Parrots: Large Language Models May Talk Causality But Are Not Causal. Transactions on Machine Learning Research.

---

> > ### Comment · Reviewer_Nzcs · 2024-11-26
> >
> > Thanks for your response. I read it. However, I'd like to maintain my score.
> >
> > Thanks.

---

### Official Review · Reviewer_L7ZX · 2024-11-03

**Soundness:** 2
**Presentation:** 3
**Contribution:** 2
**Rating:** 3
**Confidence:** 4

**Summary:**

This paper studies causal discovery by using large language models to extract the causal graph from textual documents by prompting and perform sequence prediction. Experiments were conducted on both synthetic and more realistic datasets.

**Strengths:**

* The problem itself is of interest to a wide audience
* The idea of applying LLM to this problem may inspire more researchers

**Weaknesses:**

My main concern is that the contribution seems less significnat compared with what authors claimed -- in particular:

1. the main methodology, discussed in Section 3, might be summed up as prompting LLM and getting the results. I do not see novel ways of applying the LLMs to this specific problem (I agree there was effort in designing prompts to get the model generate a JSON-formated causal graph, but that seems not to be a significnat contribution), or insights/evidence of how this approach is better and in what ways.

2. There are assumptions needed for equation (1) to work and I don't see mentions/discussions on their validaity in this setup, nor do I see a clear definition on the variables used throught section 3. For example, in the unnumbered introduction paragraph of Section 3, I understand that the causal variables/nodes refer to as noun pharases while in Section 3.2 the main focus was to adjust the conditional probability of predicted probability of tokens. There is no discussion on how one is related to the other; for example, what do you try to quantify/estimate from $P(Y|do(x),x',y')$? How does it relate to say the example in the unnumbered introduction paragraph of Section 1 ("travel restrictions diminishes airlines companies revenues")?

3. Although I appreciate the authors' efforst in doing a relative thorough experimental studies using various LLMs. I do think the results warant more discussions. For example, in Table 1, it was noted that "Only extracted answers are shown," how many exactractions failed? Does it introduce systemic bias in the evaluation?

**Questions:**

In addition to several questions I asked in the Weaknesses section --

1. Can you elaborate what variables are being intervenes and what causal effects are being studied in Section 3.2 when you write $P(Y_0|C)$ and how does this (the conditional probability of an output token $Y_0$) relates to the causal question you are trying to study?

2. In Section 3.2, the authors wrote " Due to their extensive training on a massive amount of data, LLMs are good estimators," I don't see why this necessarily implies they are also good estimators when you replace the contexts (tokens) by causal variables (say $x'$). Can you provide more details?

---

> ### Author Response · Authors · 2024-11-22
>
> Thank you for your detailled review!
>
> Regarding your questions in the weaknesses section:
>
> 1. The main contribution of our work is the use of the causal graph to perform counterfactual causal inference. Once we obtain the causal graph, we use it to perform the three steps of counterfactual inference (abduction, intervention and prediction) separately. Compared with direct counterfactual inference, this framework allows us to study more deeply where LLMs fail at this task. Moreover, we can study the ability of LLMs to make predictions on quantities based only on their causal parents.
>
> 2. Equation 1 requires the variables X,Y and U to be connected in the causal DAG and U to be the set of exogenous factors. Regarding the connection between the variables in the introduction of Section 3 and Section 3.2: the causal variables considered are subsets of the tokens in the input context $\mathbf{C}$. These subsets correspond to tokens describing the causal parents $\mathbf{pa}(Y)$ of the quantity $Y$ computed. Thank you for pointing this confusion, we will correct it in the next version of the paper.
>
> 3. The full set of answers is provided in Figure 4. We did not find any correlations between the failure cases and the type of questions. The failed extractions seem to come from the LLMs' limitations when tasked to output formatted answers and are not linked to specific input examples, so there is no bias introduced in the evaluation.
>
> Regarding your other questions:
>
> 1. The query $P(Y_0|C)$ is used as introductory example to show the difference with computing causal quantities. This query predicts a single token from the input context (it corresponds to a single forward pass with an LLM). However, predicting the correct distribution of a sequence of tokens $\mathbf{Y}$ is more challenging as it requires performing multiple predictions autoregressively. Biases and spurious correlations in the input context can alter performance during the prediction. A causal variable is a high-level concept and its description (or its instantiation) typically needs multiple tokens. To improve prediction, we consider selecting the smaller subset $\mathbf{pa}(Y) \in \mathbf{C}$ instead of the entire context. We use the extracted causal knowledge to this end.
>
> 2. You are correct! We do not imply that LLMs are good estimators of causal quantities but instead study if they can be made good estimators. We restrict the LLMs' contexts to causal parent variables and observe if they can perform direct causal inference in counterfactual settings. We hope our explanations have cleared up any confusion you may have had about the paper.

---

> > ### Comment · Reviewer_L7ZX · 2024-11-28
> >
> > Thank you for responding to my questions and for the clarifications.
> >
> > >  Regarding the connection between the variables in the introduction of Section 3 and Section 3.2: the causal variables considered are subsets of the tokens in the input context . These subsets correspond to tokens describing the causal parents of the quantity computed.
> >
> > I do think this does require a nontrivial amount of work to properly define "subsets of the tokens" as the "causal" variables and formulate the causal inference problem with them. As this is perhaps one of the building blocks of the paper, I'd like to recommend the authors to revise the paper and will keep my score.

---

### Official Review · Reviewer_TnB7 · 2024-11-04

**Soundness:** 4
**Presentation:** 3
**Contribution:** 1
**Rating:** 3
**Confidence:** 5

**Summary:**

The paper studies timely and important goal: how do we get AI systems to reason causally over natural text? They extend LLM results on inferring causal graphs to natural text datasets and perform counterfactual inference.

**Strengths:**

* The motivation of the problem and the datasets chosen are excellent.
* Experiments are clearly described and ablations are provided

**Weaknesses:**

My brief review is that the authors have done a great job setting up the problem, but their results are underwhelming.
* The proposed method introduces a lot of complex parts, but is unable to outperform the causalCoT baseline from Jin et al.
* I do not get any new insight from the results, especially because all the validation is also done by LLMs.
* While the idea is interesting, the paper's results fail to convince me that it works in practice. At least, we should see stronger results on the synthetic cladder benchmark.

**Questions:**

I have some questions below. I also provide feedback to improve the work, which the authors need not respond to.

Questions
* Is the graph generated per sentence? Or per paragraph?
* How are two graphs merged? Do you also use an LLM for this task? How do you handle variable name mismatches, mismatches in edge direction, etc,?
* How does the work relate to LLMs+causal representation learning? E.g.,this paper https://arxiv.org/abs/2402.03941
* Section 3.1 states that hidden variables are also extracted from LLM. How are they used in the causal analysis? In a dataset like cladder, does the LLM output hidden variables?
* The decision to simply ignore any questions where the LLM outputs an incorrectly formatted answer or other error is not fair. Can you present results on the entire cladder dataset? If the LLM failed to produce a valid graph, then it should be counted as a failure, since it is part of the proposed method.
* Can you think of any other validation apart from LLM self-validation? It's unclear what that evaluation is adding.

Suggestions to improve:
* I would suggest the authors to focus on cladder and show a demonstrable gain. Otherwise, it is even harder to interpret the real-world oil dataset results.
* I suspect that your method is trusting the LLM too much. Instead, once the graph is obtained, we can do many operations symbolically and invoke LLMs only for small, focused tasks. So I would encourage you to look into a direction where the central control is handled by the SCM and CF procedure and the modular atomic tasks are delegated to LLMs. Some iteration of this idea should easily give higher accuracies on cladder, assuming that the graph is correct.

---

> ### Author Response · Authors · 2024-11-22
>
> Thank you for your review and for the suggestions! We understand that the proposed model does not outperform other baselines. However, our aim with this work is to give insight into the current abilities and limitations of LLMs when performing counterfactual inference. By dividing every step into atomic components, we show that their primary limitation comes from prediction errors and not abduction or intervention errors. We think that these findings and our proposed directions can help drive research in the domain.
>
> Regarding your questions:
>
> 1. The graph is generated per paragraph. We task the LLM to read read the text and return the causal relationships explicitly written in the text.
>
> 2. Graph merging is indeed a challenging problem, sometimes referred as the structural causal marginal problem [1]. As it is an open problem, we use a simple solution to be efficient. We build node embeddings to find the same variables in different graphs and use a clustering algorithm to merge them. We do no merge nodes if this leads to a cycle in the graph. We provide additional details on graph merging in Appendix A.
>
> 3. The literature on causal structure discovery and causal representation learning focuses on recovering causal relationships that exist **implicitly** in the data (e.g. if a drug has an effect on a disease by reading patient data). This is a very challenging problem that typically involves performing conditional independence tests between the variables [2]. Our work focuses on latter stages in the causal inference pipeline: we use LLMs to find causal relationships **explicitly** stated in natural language sources and study counterfactual inference from the built causal graphs.
>
> 4. We prompt the LLM to propose causal variables not explicitly stated in the data but can plausibly affect the observed variables. This step relies on commonsense causal reasoning, an task LLMs have been shown to be good at [3,4]. These variables do not have an instantiated value as they are unobserved. In the abduction step, we retrieve their values during the abduction step via inference in the anticausal direction (i.e. by computing P(U|children(U)). The found values are then used during intervention and prediction. In cladder, all the variables have an instantiation (even the ones named "confounder"), so there are no hidden variables.
>
> 5. This is a good point! We decided to discard wrongly formatted answers in Table 1 to ease interpretation. This table compares LLMs' abilities to perform counterfactual inference. Our method introduces additional sources of errors that are not present with direct prompting of chain-of-thought, including them would have made harder to compare the LLMs on the reasoning parts. We understand that it means that we are not directly comparing our method but we favoured this choice as our aim is to better understand the limitations of LLMs in counterfactual reasoning. We also include the full results with parsing failures in Figure 4. We will work on reducing parsing errors to eliminate this problem.
>
> 6. The self-evaluation is used for assessing if the LLMs can detect wrong or unconfident answers. If close to ground-truth, self-evaluation could be used in real-world cases where the ground-truth is not available. We will add an in-depth comparison between accuracy and self-evaluation in the next version of the paper to improve this section.
>
>
> [1] Gresele, L., Von Kügelgen, J., Kübler, J., Kirschbaum, E., Schölkopf, B., & Janzing, D. (2022, June). Causal inference through the structural causal marginal problem. In International Conference on Machine Learning (pp. 7793-7824). PMLR.
>
> [2] Judea Pearl. Causality. 2nd ed. Cambridge university press, 2009. doi: 10.1017/CBO9780511803161.
>
> [3] Kıcıman, E., Ness, R., Sharma, A., & Tan, C. (2023). Causal reasoning and large language models: Opening a new frontier for causality. arXiv preprint arXiv:2305.00050.
>
> [4] Zhang, J., Zhang, H., Su, W., & Roth, D. (2022, June). Rock: Causal inference principles for reasoning about commonsense causality. In International Conference on Machine Learning (pp. 26750-26771). PMLR.

---

> > ### Comment · Reviewer_TnB7 · 2024-11-26
> > **thank you for your response!**
> >
> > I appreciate the detailed response. It helped me understand some of the details better. However, I still feel that the paper needs one more round of work; so I will keep my score.

---

### Official Review · Reviewer_v566 · 2024-11-05

**Soundness:** 3
**Presentation:** 4
**Contribution:** 3
**Rating:** 6
**Confidence:** 4

**Summary:**

The paper proposes an approach for causal discovery and counterfactual inference from unstructured natural language. The method uses LLMs to extract causal variables, build causal graphs, and conduct counterfactual reasoning based on these inferred structures. The authors introduce a two-step framework: first, they prompt the LLM to generate causal graphs by identifying relationships between events in the text, potentially merging graphs from various documents for broader coverage. Then, the inferred graphs are used to simulate counterfactual scenarios, allowing them to analyze and evaluate causal influences with an LLM. The authors conduct experiments on synthetic and real-world datasets to illustrate the model's ability to handle counterfactual queries and identify sources of inference limitations, primarily prediction errors in reasoning tasks. Through these experiments, the paper highlights limitations in current LLM capabilities regarding counterfactual reasoning and suggests that prediction errors are a significant bottleneck.

**Strengths:**

The main strength of the paper is in exploring how LLMs can support causal reasoning tasks on text data, pointing to new possibilities and practical challenges. The authors introduce a novel approach for discovering causal structures and conducting counterfactual inference directly from unstructured text using LLMs.

The methodology is well-defined, with clear steps in building and merging the causal graphs.

The experiments, both on synthetic and on real data, demonstrate the applicability of the framework.

**Weaknesses:**

1. The paper’s reliance on synthetic data from Cladder restricts its assessment of model performance in realistic contexts. The model is only demonstrated on a few real-world news examples, which doesn’t fully establish its applicability to unstructured text data beyond controlled scenarios.

2. Expanding the analysis to include diverse domains would provide a stronger basis for understanding the versatility and potential challenges fro the LLM-based approach.

3. The paper could benefit from quantitative performance metrics on real-world data (e.g., precision and recall for causal variable extraction, fidelity of counterfactual predictions).

4. The paper would be improved by addressing issues like hallucination and prompt sensitivity, as these can impact the accuracy of causal inference.

**Questions:**

1. Could you provide additional quantitative evaluations on real-world datasets beyond the limited examples in the paper? Specifically, metrics like precision, recall, or causal graph fidelity in real news articles or other domains.

2. Given the reliance on the synthetic Cladder data, do you see limitations in how well this dataset represents real-world causal complexity?

3. How do you think the approach handles potential issues with LLM hallucination or prompt sensitivity?

4. The paper mentions that counterfactual inference accuracy is a primary limitation, largely due to prediction errors. Could you provide more details on specific failure cases?

---

> ### Author Response · Authors · 2024-11-22
>
> Thank you for your detailed review and for the advice on how to improve the paper!
>
>
> Regarding your questions:
>
> 1. Yes, this is a direction we are interested in pursuing. We have conducted separate experiments to quantify the quality of the LLM extractions. In this set of experiments, we prioritized getting a fully automated quality assessment. To that end, we considered extracting wiki concepts from the causal variables described in the nodes and the edges of our causal graphs and computed the Jaccard similarity between them. Analyzing nearly 170 causal graphs, we found that the Jaccard similarity was zero for about 55% of the cases, while for only ~10% of the cases, we observed a perfect match. When considering the results, we must note that the Jaccard similarity does not consider the semantic proximity of the wiki concepts: we defer such an evaluation to future work. This evaluation shows that different kind of information is retained in the nodes encoding the causal variables, and the edges informing about the causal relationships. Future work will focus on aligning them better. Furthermore, we will work toward building and releasing a golden dataset against which we could measure the quality of the extracted causal relationships against human-annotated ground truth for causal graphs extracted from media news. To complement this view with a counterfactual perspective, we would like to underscore that conducting counterfactual causal inference on real-world data is challenging. In most settings, we can only observe one factual world and do not have access to possible counterfactual ground truths (this is the fundamental problem of causal inference [1]). Due to this limitation, there are very few existing datasets that could be used for this task and the ones that exist may be solvable via commonsense reasoning and not true causal reasoning [2]. Nevertheless, we are already looking into ways to overcome this challenge. Part of our future research will focus on finding or building datasets to alleviate these shortcomings.
>
> 2. You raise an important point: built synthetic causal structures can differ from real-world causal structures. We highlight this difference by comparing the causal graphs in cladder with the ones extracted from real-world news articles. The main differences observed are the size of the causal structures and the account of possible missing variables: real-world causal structures are bigger and more complex than synthetic ones and can contain hidden confounders.
>
> 3. We try different prompting strategies to ensure we are not obtaining suboptimal results due to the prompt. Our approach directly aims to reduce the risks of hallucinations during counterfactual inference by reducing the context size to the direct causal parents of the quantity to compute, thereby reducing the number of context tokens generated and the risks of hallucination [3]. Furthermore, we are now exploring additional refinements that would help us to (a) validate and correct the data types assigned to the causal variables and (b) leverage techniques that would allow numerically confirm the causal relationship between them and their direction. This would solve an open research question and a gap that currently could not be solved with the LLMs alone [4].
>
> 4. Yes, we provide an example in Section 4.4. In this example, the inference to perform is a logical OR between two parent variables. This information is directly extracted from the input text and included in the prompt. However, GPT-4o interprets it as logical AND (we know this from the chain-of-thought explanation given by the model). The model returns a wrong answer due to an error during the prediction step, while the abduction and intervention steps were correct. Therefore, the error is a failure to process a logical entailment and is not due to a causal reasoning error.
>
>
> [1] Judea Pearl. Causality. 2nd ed. Cambridge university press, 2009. doi: 10.1017/CBO9780511803161.
>
> [2] Zečević, M., Willig, M., Dhami, D. S., & Kersting, K. Causal Parrots: Large Language Models May Talk Causality But Are Not Causal. Transactions on Machine Learning Research.
>
> [3] Fadeeva, E., Vashurin, R., Tsvigun, A., Vazhentsev, A., Petrakov, S., Fedyanin, K., ... & Shelmanov, A. (2023). LM-polygraph: Uncertainty estimation for language models. arXiv preprint arXiv:2311.07383.
>
> [4] Joshi, N., Saparov, A., Wang, Y., & He, H. (2024). LLMs Are Prone to Fallacies in Causal Inference. arXiv preprint arXiv:2406.12158.

---

### Meta-Review · Area_Chair_RKeK · 2024-12-21

**Metareview:**

The paper proposes a two-step approach that first extracts causal graphs from text and then performs counterfactual reasoning on these graphs.

Strengths:

+ Studies an important and timely problem in AI research with potential practical applications

Weaknesses:

+ Heavy reliance on synthetic data (Cladder dataset) with limited real-world validation

+ Unable to outperform existing baselines like causalCoT, with underwhelming results particularly on benchmark tasks

+ Lack of quantitative evaluation metrics for real-world performance (e.g., precision/recall) and insufficient handling of LLM-specific issues like hallucination and prompt sensitivity

**Additional Comments On Reviewer Discussion:**

The reviewers are in agreement that the paper are not ready for publication in its current form.

---

### Decision · Program_Chairs · 2025-01-22

Reject